Brief Communication

# Limits to the accurate and generalizable use of soundscapes to monitor biodiversity

**Sarab S. Sethi** [1,2] ✉, **Avery Bick**[3], **Robert M. Ewers** [4], **Holger Klinck**[5], **Vijay Ramesh**[5,6], **Mao-Ning Tuanmu** [7] & **David A. Coomes** [1]

Although eco-acoustic monitoring has the potential to deliver biodiversity insight on vast scales, existing analytical approaches behave unpredictably across studies. We collated 8,023 audio recordings with paired manual avifaunal point counts to investigate whether soundscapes could be used to monitor biodiversity across diverse ecosystems. We found that neither univariate indices nor machine learning models were predictive of species richness across datasets but soundscape change was consistently indicative of community change. Our findings indicate that there are no common features of biodiverse soundscapes and that soundscape monitoring should be used cautiously and in conjunction with more reliable in-person ecological surveys.

Anthropogenic pressures are impacting biodiversity globally[1]. Declines in species richness catch headlines[2] but changes in community composition can have just as devastating ecological effects[3]. To design evidence-based conservation measures, monitoring biodiversity is essential and listening to the sounds produced by an ecosystem (acoustic monitoring) holds promise as a scalable and inexpensive way to achieving this[4].

Many species contribute to an ecosystem's soundscape, whether through producing vocalizations (for example, birdsong), stridulations (for example, cricket chirps) or as they interact with the environment (for example, the buzz of a bee). Streaming[5] or recording[6] soundscapes across huge scales is now common[7–9], yet interpreting the audio to derive biodiversity insight remains a challenge. Automating the identification of stereotyped sounds in audio recordings[10] can provide species occurrence data on large scales, building a bottom-up picture of biodiversity. However, vocalization detection algorithms rely on vast amounts of training data[11], meaning even state-of-the-art models are only able to reliably detect vocalizations from the most commonly found species[12]. An alternative top-down approach is to use the features of an entire soundscape to infer biodiversity[13]. Entropy-based acoustic indices[13] or embeddings from machine learning models[14] have both been used to predict community richness or ecosystem intactness with some success when calibrated using trusted independent sources of ground-truth biodiversity data (hereafter, ground-truth data)[15,16]. Nonetheless, soundscape features which correlate positively with biodiversity at one site can have an inverse relationship in another[17,18] and no reliable single metric has been found[19]. The inability of both vocalization detection and soundscape approaches to generalize has meant that acoustic monitoring has only provided ecological insight in already well-studied regions and the technology's transformative potential has remained largely unfulfilled.

We collated 8,023 short (1–20 min) soundscape recordings collected concurrently with manually recorded avifaunal community data from four diverse datasets: a temperate forest in Ithaca, USA ($n$ = 6,734, one site), a varied tropical rainforest landscape in Sabah, Malaysia ($n$ = 977, 14 sites), an agricultural tea landscape in Chiayi, Taiwan ($n$ = 165, 16 sites) and a varied montane tropical forest and grassland landscape in the Western Ghats, India ($n$ = 147, 91 sites). In India, Malaysia and Taiwan, avian community data were collected by in-person point counts performed by experts, whereas citizen science checklists from eBird were used for the USA[20] (Methods). For each recording, we calculated two common types of acoustic features: (1) a 128-dimensional convolutional neural network (CNN) embedding (learned features, LFs)[14,21] and (2) 60 analytically derived soundscape indices (SSIs)[22].

[1]Conservation Research Institute and Department of Plant Sciences, University of Cambridge, Cambridge, UK. [2]Centre for Biodiversity and Environment Research, University College London, London, UK. [3]Norwegian Institute for Nature Research, Trondheim, Norway. [4]Georgina Mace Centre for the Living Planet, Department of Life Sciences, Imperial College London, London, UK. [5]K Lisa Yang Center for Conservation Bioacoustics, Cornell University, Ithaca, NY, USA. [6]Project Dhvani, Bangalore, India. [7]Biodiversity Research Center, Academia Sinica, Taipei, Taiwan. ✉e-mail: sss70@cam.ac.uk

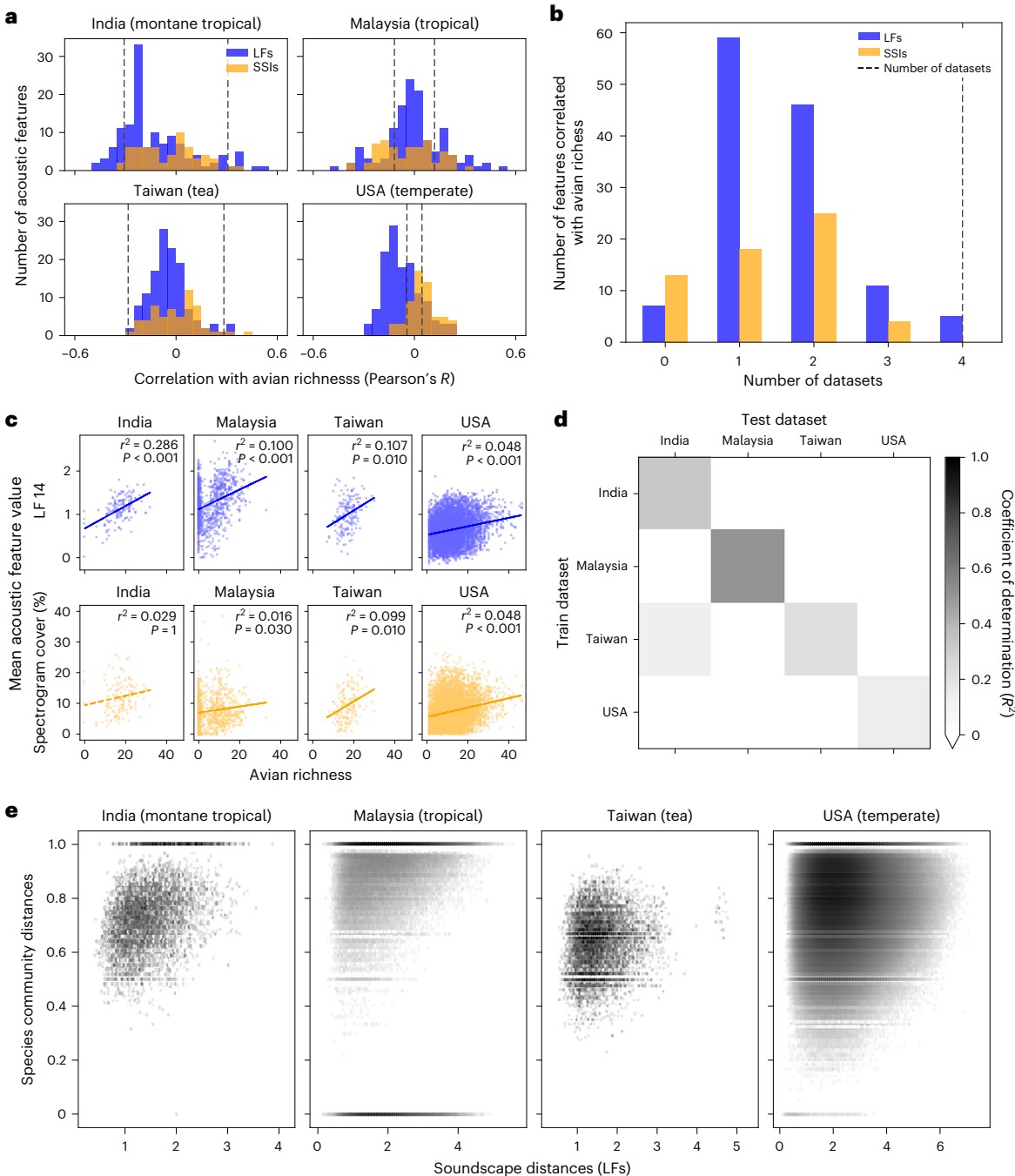

**Fig. 1 | Predictions of avian species richness from acoustic features of soundscapes are not generalizable across datasets but soundscape change is a reliable indicator of community change. a**, Many SSIs and LFs correlated significantly with avian species richness in each dataset. Dashed lines indicate significance thresholds derived from two-sided permutation tests and overlaps between the SSI (orange) and LF (blue) histograms are in brown. **b**, However, only four LFs and no SSIs correlated significantly with richness in all datasets, with most only correlating with richness in one or two datasets (105 LFs and 43 SSIs). **c**, For spectrogram cover and LF 14 (both of which correlated with richness in many datasets with two-sided Pearson's tests) the gradient and intercept of the correlations varied between datasets and correlation coefficients were relatively low, limiting transferability. **d**, Predicting avian richness with a machine learning model trained on LFs was moderately successful when training and test datasets were the same. However, models were unable to generalize when predicting richness in datasets other than the one used for training. **e**, For each dataset we found that pairwise distances between the acoustic features of soundscapes correlated with Jaccard distances between avian communities present. With Jaccard distances, 0 indicates identical communities and 1 indicates no community overlap. Number of samples per dataset: India ($n = 147$), Malaysia ($n = 977$), Taiwan ($n = 165$) and the USA ($n = 6,734$).

First, we investigated univariate correlations between acoustic features and avian species richness within each dataset. We found that many LFs and SSIs correlated significantly with richness (Pearson's correlation $P < 0.05$ by permutation test with Bonferroni correction), although the number varied greatly by dataset (Fig. 1a). Most acoustic features correlated significantly with species richness in either one or two of the four datasets (105 LFs, 43 SSIs; Fig. 1b). Only four LFs and not a single SSI correlated with avian richness across all four datasets (Fig. 1b). For features that were correlated with avian richness in multiple datasets, the gradient and intercept of fitted lines varied and

correlation coefficients were relatively low (for example, 14th learned feature: $0.048 < r^2 < 0.286$, ROICover (a measure of spectrogram cover): $0.002 < r^2 < 0.099$, Fig. 1c).

By training a machine learning model on the full-dimensional learned feature vectors, we were able to predict species richness within datasets with relative success (coefficient of determination $R^2 = 0.32$, 0.50, 0.22 and 0.14 for India, Malaysia, Taiwan and the USA, respectively; Fig. 1d). However, except for in one case (trained on Taiwan, tested on India, $R^2 = 0.13$), the models were unable to predict richness when evaluated on datasets that they were not trained upon ($R^2 < 0$, Fig. 1d). There was no significant correlation between dataset sample sizes and within-dataset or mean cross-dataset $R^2$ values (Pearson's correlation $P > 0.05$). Similar results were found when using only the 60 SSIs or all 188 features (128 LFs + 60 SSIs) together to predict richness (Extended Data Fig. 1). Our results indicate that even within a single dataset—but especially when looking across datasets—soundscapes with similar levels of avian diversity do not share similar acoustic features.

Rather than attempting to predict species richness directly, we investigated the relationship between pairwise Euclidean distances between the mean acoustic features of each audio recording (soundscape change) and pairwise Jaccard distances between their associated avian species communities (community change). We found strong significant correlations between soundscape change and community change within each dataset (Spearman correlation Mantel test, $P \leq 0.001$ for all; Fig. 1e). There were more examples where communities changed but soundscapes did not than the converse, indicated by clustering of points in the upper left of Fig. 1e. No significant correlations were found between soundscape change and change in species richness in any of the four datasets (Extended Data Fig. 2). For each of the 14 sites in Malaysia—the only dataset with sufficient replicates at individual sites to test for this relationship—there was a significant correlation between soundscape change and community change ($P \leq 0.03$). The decision to use LF (Fig. 1e) or SSIs (Extended Data Fig. 3) to track soundscape change did not alter our findings.

In all four datasets, we found that certain acoustic features correlated with avian richness. However, features did not behave consistently across datasets and most were only correlated with richness in one or two datasets (explaining inconsistencies seen in the literature[19]; Fig. 1a–c). Without access to independent sources of trusted ground-truth biodiversity data, ascertaining which features were most suitable for each dataset would be impossible, limiting the transferability of this approach. We then found that a machine learning model trained on compound indices was able to produce within-dataset predictions of species richness with some success. However, again, generalizability was not achieved as models did not produce informative estimates when applied to datasets they were not trained upon (Fig. 1d). The diversity of flora, fauna, survey designs and recording equipment across the datasets might, perhaps, make these results unsurprising. Indeed, further studies may show that limited generalizability can be achieved when study biomes and survey methodologies are closely matched. Nevertheless, given the current state-of-the-art, our results stress that however well an acoustic feature or machine learning model may perform in one scenario, without access to high-quality ground-truth data from the exact location being studied, soundscape methods should not be used to generate predictions of species richness.

In contrast to the unpredictable behaviour of the models producing estimates of avian richness, we found that soundscape change correlated with community change in all datasets (Fig. 1e). However, this approach was also limited, as there were many examples where similar soundscapes were associated with very different avian communities. One reason could be that non-biotic sounds (for example, motors) or non-avian biotic sounds (for example, insects and amphibians) were larger contributors to soundscapes than were birds[15,23]. Our ability to predict avian richness was therefore more likely to have been based on latent variables measuring habitat suitability (for example, vocalizing prey or nearby water sources) rather than being driven by the vocal contributions of birds directly.

Despite having access to vast amounts of manually collected avifaunal point count data, even within datasets, soundscape predictions of avian richness and community change struggled with accuracy. This result suggests that even if we can collate standardized global soundscape datasets and train models using ground-truth data from every biome on earth, we may still be left wanting. Rather than relying on automated predictions in isolation, a more reliable approach might be to use soundscapes to collect coarse but scalable data, which can be used to direct more detailed ecological studies. For example, data from large-scale acoustic monitoring networks could be used to direct in-person surveys towards only the sites with unexpected and prolonged soundscape changes or large shifts in predicted biodiversity. Such an approach could result in more efficient use of limited expert resources to ensure that conservation interventions are deployed in a timely and focussed manner. Indeed, due to issues surrounding interpretability and accountability, we will probably always require expert verification of autonomous monitoring outputs before policy and management practices are modified[24]. Proceeding with more realistic expectations around how soundscapes can best contribute large-scale biodiversity monitoring efforts will be essential to maximizing the transformative potential of the technology.

## Methods

### Avifaunal point counts

**India.** Data were collected from 91 sites in the Western Ghats between March 2020 and May 2021. Sites were a mixture of montane wet evergreen and semi-evergreen forests (29), montane grasslands (8), moist deciduous forests (7), timber plantations (17), tea plantations (20), agricultural land (8) and settlements (2). Sites had a mean separation distance of 18.9 km, with the closest two being 824 m apart.

In total, 147 15-minute point counts were conducted (mean 1.6 per site). All point counts were conducted between 06:00 and 10:00 as these were the hours with highest expected avian activity. A variable-distance point count approach was followed and all bird species heard, seen and those that flew over (primarily raptor species) were noted. A total of 119 avifaunal species were recorded across all point counts.

Single channel audio was recorded during each point count using an Audiomoth device raised 1–2 m from the ground[6]. Recordings were saved in WAV format at a sampling rate of 48 kHz.

**Malaysia.** Data were collected from a varied tropical landscape at the Stability of Altered Forest Ecosystems (SAFE) project[25] in Sabah between March 2018 and February 2020. The 14 sites spanned a degradation gradient: two in protected old growth forest, two in a protected riparian reserve, six in selectively logged forest (logging events in 1970s and early 2000s), two in salvage logged forest (last logged in early 2010s) and two in oil palm plantations. Sites had a mean separation distance of 7.6 km, with the closest two being 583 m apart.

In total, 977 20-minute avifaunal point counts were performed across 24 hours of the day (59–80 per site). During point counts, all visual or aural encounters of avifaunal species within a 10 m radius of the sampling site were recorded. Species identifications and names were validated using the Global Biodiversity Information Facility[26]. A total of 216 avifaunal species were recorded across all point counts.

Single channel audio was recorded during each point count with a Tascam DR-05 recorder mounted to a tripod raised 1–2 m from the ground (integrated omnidirectional microphone, nominal input level −20 dBV, range 20 Hz–22 kHz). Recordings were saved in WAV format at a sampling rate of 44.1 kHz.

**Taiwan.** Data were collected from 16 tea plantations in the Alishan tea district, located in Chiayi County of Taiwan, between January and November 2022. The tea plantations spanned an elevation gradient from 816 to 1,464 m and were surrounded by secondary broadleaf forests or coniferous plantations.

In total, 176 10-minute avifaunal point counts were conducted, on average once per site per month. Every survey was conducted within 3 hours after sunrise. During point counts, the species of every bird individual visually or aurally detected was recorded and its horizontal distance from the observer was estimated as 0–25, 25–50, 50–100, >100 m or flying over. Our process for matching audio recordings to point counts is provided later in the Methods. A total of 81 avifaunal species were recorded across all point counts used in this study.

Stereo audio was recorded at a sampling rate of 44.1 kHz for one of every 15 minutes at each site throughout the sampling period. Wildlife Acoustics Song Meter 4 devices were used (mounted to a tree trunk or tripod raised 1–2 m from the ground) and data were saved in WAV format.

**The USA.** Data were collected from a single site in a temperate forest at Sapsucker Woods, Ithaca, USA, from January 2016 to December 2021.

In the absence of standardized point counts across such a long duration, we used eBird checklists to determine avian communities (only possible since Sapsucker Woods is a major hotspot for eBird data). Data were filtered to only keep checklists which were complete, of the 'travelling' or 'stationary' types, located within 200 m of the recording location and lasted less than 30 minutes (to minimize bias from varying sampling intensities). We combined occurrence data from all eBird checklists that started during each audio recording to create one pseudo point count per audio file. Since we only use occurrence data, double counting of the same species was not an issue. This resulted in a total of 6,734 point counts and 231 species.

Audio was recorded continuously throughout the sampling period using an omnidirectional Gras-41AC precision microphone raised approximately 1.5 m from the ground and digitized with a Barix Instreamer ADC at a sampling rate of 48 kHz. Files were saved in 15-minute chunks and in WAV format.

### Avian communities

For each point count, we used the field observations to derive community occurrence data (that is, presence or absence) since not all datasets had reliable abundance information. To calculate avian richness, we calculated the unique number of species encountered within each point count ($\alpha$-diversity).

### Acoustic features

We used two approaches to calculate acoustic features for each recording. The first was using a suite of 60 SSIs, using the scikit-maad library (v.1.3)[22]. Specifically, we appended together the feature vectors from the functions maad.features.all_temporal_alpha_indices and maad.features.all_spectral_alpha_indices. In all cases, we used default arguments for generating features, since tuning parameters to each of the datasets would have been a cumbersome and inherently qualitative process susceptible to introducing bias. A full list of features is given in Supplementary Table 1.

The second approach was to use a learned feature embedding LF from the VGGish CNN model[21]. VGGish is a pretrained, general-purpose sound classification model that was trained on the AudioSet database[27]. First, to transform raw audio into the input format expected by the pretrained VGGish model[21], audio was downsampled to 16 kHz, then converted into a log-scaled mel-frequency spectrogram (window size 25 ms, window hop 10 ms, periodic Hann window). Frequencies were mapped to 64 mel-frequency bins between 125 and 7,500 Hz and magnitudes were offset before taking the logarithm. Data were inputted to the CNN in spectrograms of dimension 96 × 64, producing one 128-dimensional acoustic feature vector for each 0.96 s of audio.

For all acoustic features (LFs and SSIs), mean feature vectors were calculated for each point count. For Malaysia, India and the USA, recordings had a 1:1 mapping with point counts, so features were averaged across each audio recording. For Taiwan, where audio data were sparser, features were averaged across all audio recordings that begun within a 1-hour window centred around the start time of the point count, resulting in an average of four 1-minute audio recordings per point count. Using window sizes of 40 or 20 minutes around the point count start time in Taiwan did not change our results (Fig. 1d reproduced with different window sizes and Extended Data Fig. 4).

### Univariate correlations with avian richness

Univariate Pearson correlations were calculated between each of the 60 SSIs and 128 LFs with avian richness across all point counts within datasets. Significance thresholds were determined by calculating 100 null correlation coefficients between shuffled features and avian richness. Null correlation coefficients were aggregated for each dataset across all 60 SSIs and 128 LFs (total null values 18,800). Accounting for a Bonferroni multiple hypothesis correction, the threshold for significance at $P = 0.05$ was taken as value 18,784 in an array of sorted absolute null correlation coefficients in ascending order. For each dataset, all real correlations between acoustic features and avian richness with an absolute correlation coefficient greater than this threshold were determined to be significant. Lines of best fit in Fig. 1c were determined by fitting a first-order polynomial to the data on avian richness (x axis) and the single acoustic feature being considered (y axis).

### Machine learning predictions of avian richness

We used a random forest regression model to make predictions of avian richness using the full-dimensional acoustic feature vectors. In Extended Data Fig. 1b, the mean of all 60 SSIs and 128 LFs were combined to create one 188-dimensional feature vector per audio recording. The scikit-learn implementation RandomForestRegressor was used with a fixed random seed and other default parameters unchanged. We randomly selected 70% of point counts in each dataset for training and held back 30% for testing. Splitting train and test data randomly represents a 'best-case' scenario since, within datasets, training data distributions will closely match test data distributions. We measured how well predicted richness matched true richness using the coefficient of determination ($R^2$). Both within and across datasets (that is, for all results in Fig. 1d and Extended Data Fig. 1), models were always trained on the training data and scores were evaluated on the test data.

We used Pearson's correlation coefficient to test whether there was a link between dataset sample sizes and classifier performance. For within-dataset performance, we used the diagonal values of Fig. 1d. For cross-dataset performance, we took the mean of each row in Fig. 1d, excluding the diagonal values.

### Investigating link between soundscape change and community change

To measure community change between two point counts we used the Jaccard distance metric, where 0 indicates no change in community and 1 indicates no shared species between communities. To measure soundscape change we calculated the Euclidean distance between the mean acoustic feature vectors for each point count. Distances were computed in the full 128-dimensional space for the LFs (Fig. 1e) and 60-dimensional space for the SSIs (Extended Data Fig. 3). Performing these two operations between all point counts within each dataset, we derived two matrices representing pairwise community change and pairwise soundscape change. Since community change was bounded between 0 and 1, a Pearson correlation was not appropriate. Therefore, the Spearman correlation between these two matrices was measured using a one-tailed Mantel test, with $P$ values derived empirically (using the skbio implementation skbio.stats.distance.mantel).

## Reporting summary

Further information on research design is available in the Nature Portfolio Reporting Summary linked to this article.

## Data availability

Processed acoustic features and avian community data for all point counts are published at https://doi.org/10.5281/zenodo.7410357.

## Code availability

Code with instructions to reproduce analyses, results and figures can be found at https://doi.org/10.5281/zenodo.7458688.

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

## Acknowledgements

We would like to thank staff at the SAFE project, the Cornell Lab of Ornithology and Ecoacoustics and Spatial Ecology Lab at Academia Sinica for their support in data collection and preprocessing. Specific thanks go to J. Sleutel, A. Shabrani, N. Zulkifli, R. Mack, B. Thomas, A. Anand, C. Das, A. Rajan, C.-Y. Lee and S.-H. Liu. This project was supported by funding from the Herchel Smith Fund (S.S.S.), World Wildlife Fund (Malaysia data), Sime Darby Foundation (Malaysia data), National Science and Technology Council (NSTC 111-2321-B-002-019 and 111-2927-I-001-513; Taiwan data) and Biodiversity Research Center at Academia Sinica (Taiwan data). Feature computations for the US dataset were performed on resources provided by Sigma2—the National Infrastructure for High Performance Computing and Data Storage in Norway. Data were collected from Malaysia under an SaBC permit granted to S.S.S. (JKM/MBS.1000-2/2 JLD.8 (63)).

## Author contributions

S.S.S. conceptualized the study with inputs from D.A.C. and R.M.E. S.S.S., R.M.E., H.K., V.R. and M.-N.T. contributed to data collection and cleaning. S.S.S. and A.B. analysed the data, with input from all authors. S.S.S. led writing, with input from all authors.

## Competing interests

The authors declare no competing interests.

## Additional information

**Extended data** is available for this paper at https://doi.org/10.1038/s41559-023-02148-z.

**Correspondence and requests for materials** should be addressed to Sarab S. Sethi.

(a) 60 SSIs

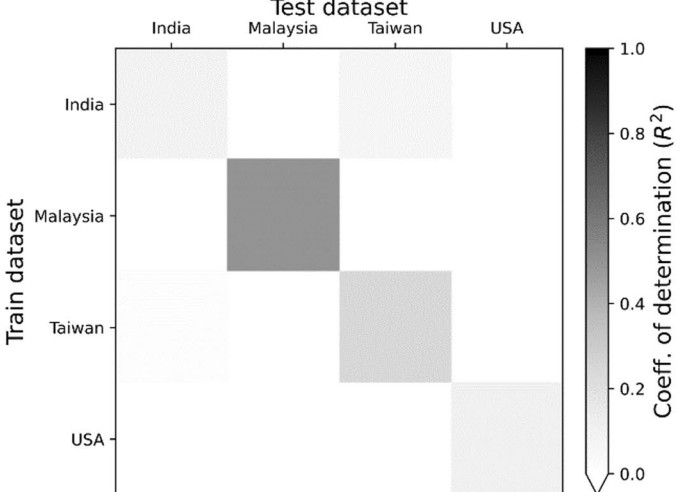

(b) All 188 features (60 SSIs + 128 LFs)

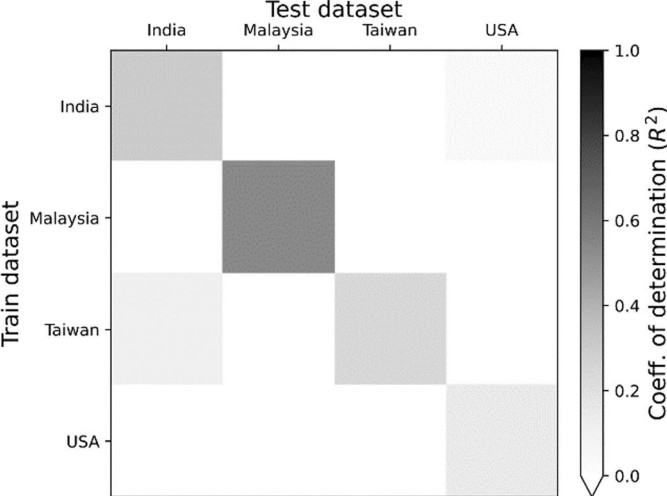

**Extended Data Fig. 1 | Predictions of species richness were not generalisable using alternative feature sets.** When training and test datasets were the same, predictions of species richness from a machine learning model (Random Forest Regressor) trained on (**a**) the 60 soundscape indices (SSIs) or (**b**) all 188 acoustic features (60 SSIs + 128 LFs) were of varying levels of accuracy (R2 = 0.09, 0.50,

0.24, 0.10 [SSIs] R2 = 0.30, 0.54, 0.24, 0.15 [all 188 features] for India, Malaysia, Taiwan, USA, respectively). In all cases, predictive models did not generalize to produce informative estimates of species richness when applied to datasets they were not trained on. Similar results using only the 128 LFs are shown in Fig. 1c in the main text.

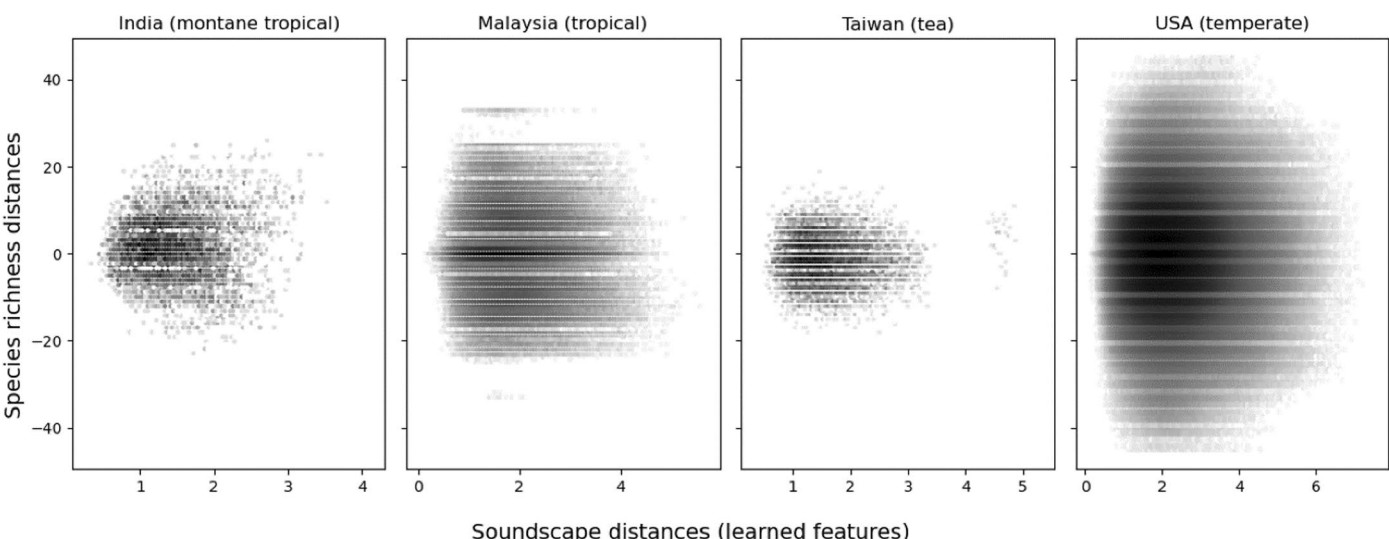

**Extended Data Fig. 2 | Soundscape change did not correlate with change in species richness.** No significant correlations were found between change in soundscape features and change in species richness in any of the four datasets (Spearman's correlation two-sided Mantel test, p > 0.05 in all cases).

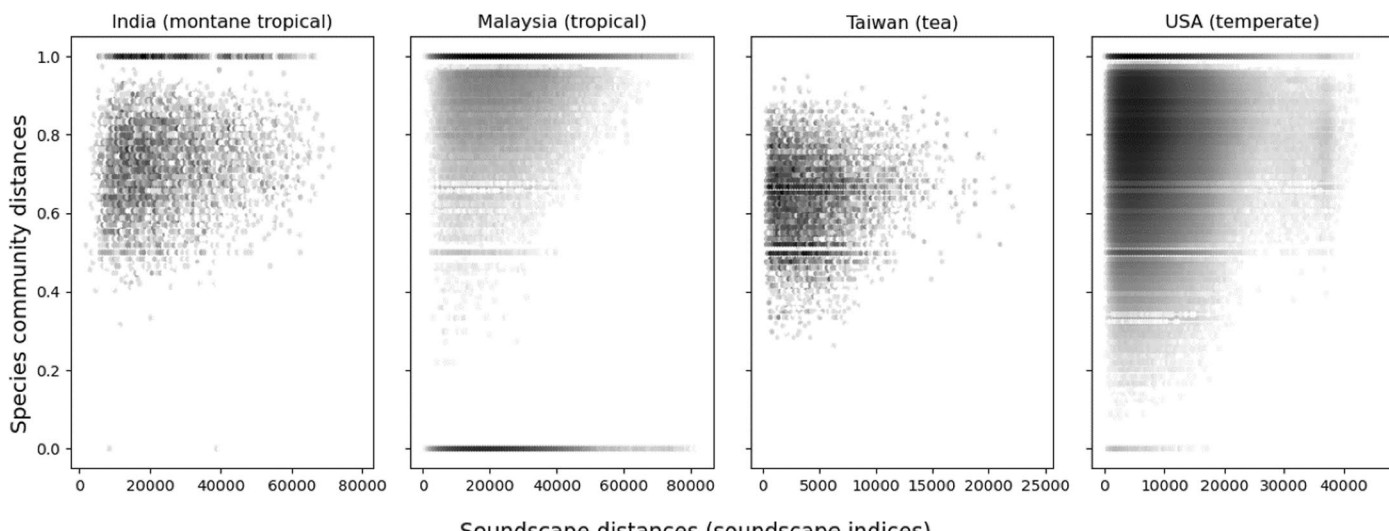

**Extended Data Fig. 3 | Soundscape change correlated with change in species community using soundscape indices.** Even when using soundscape indices rather than learned features, change in acoustic features was correlated with change in avian community across all datasets (Spearman's correlation two-sided Mantel test, p = 0.002, 0.001, 0.023, 0.001 for India, Malaysia, Taiwan, USA).

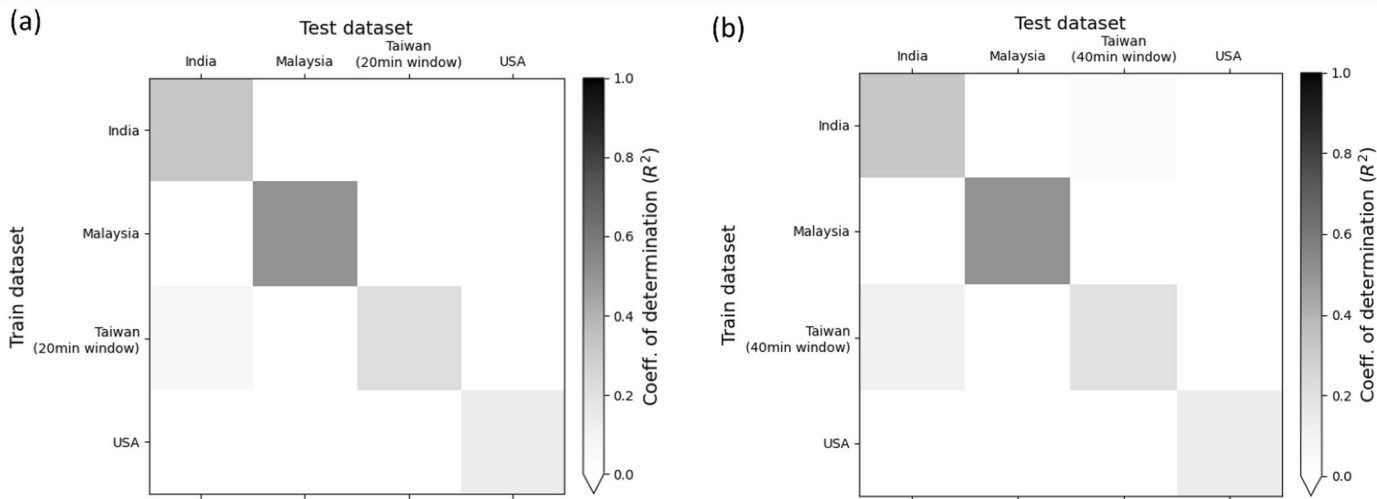

**Extended Data Fig. 4 | Window size for the Taiwan dataset did not impact our results.** The chosen size of the window around the point count start time in the Taiwan dataset did not change the accuracy or generalizability of the machine learning models trained to predict species richness.

# Reporting Summary

## Statistics

For all statistical analyses, confirm that the following items are present in the figure legend, table legend, main text, or Methods section.

| n/a | Confirmed | |
|---|---|---|
| ☐ | ☒ | The exact sample size (*n*) for each experimental group/condition, given as a discrete number and unit of measurement |
| ☐ | ☒ | A statement on whether measurements were taken from distinct samples or whether the same sample was measured repeatedly |
| ☐ | ☒ | The statistical test(s) used AND whether they are one- or two-sided *Only common tests should be described solely by name; describe more complex techniques in the Methods section.* |
| ☒ | ☐ | A description of all covariates tested |
| ☐ | ☒ | A description of any assumptions or corrections, such as tests of normality and adjustment for multiple comparisons |
| ☐ | ☒ | A full description of the statistical parameters including central tendency (e.g. means) or other basic estimates (e.g. regression coefficient) AND variation (e.g. standard deviation) or associated estimates of uncertainty (e.g. confidence intervals) |
| ☐ | ☒ | For null hypothesis testing, the test statistic (e.g. *F*, *t*, *r*) with confidence intervals, effect sizes, degrees of freedom and *P* value noted *Give P values as exact values whenever suitable.* |
| ☒ | ☐ | For Bayesian analysis, information on the choice of priors and Markov chain Monte Carlo settings |
| ☒ | ☐ | For hierarchical and complex designs, identification of the appropriate level for tests and full reporting of outcomes |
| ☐ | ☒ | Estimates of effect sizes (e.g. Cohen's *d*, Pearson's *r*), indicating how they were calculated |

*Our web collection on statistics for biologists contains articles on many of the points above.*

## Software and code

Policy information about availability of computer code

| Data collection | No software was used for data collection |
|---|---|
| Data analysis | Software for analysis can be found at https://github.com/sarabsethi/sscape-avian-div-generalisability (versioned at https://zenodo.org/record/7845899) |

For manuscripts utilizing custom algorithms or software that are central to the research but not yet described in published literature, software must be made available to editors and reviewers. We strongly encourage code deposition in a community repository (e.g. GitHub). See the Nature Portfolio guidelines for submitting code & software for further information.

## Data

Policy information about availability of data

All manuscripts must include a data availability statement. This statement should provide the following information, where applicable:
- Accession codes, unique identifiers, or web links for publicly available datasets
- A description of any restrictions on data availability
- For clinical datasets or third party data, please ensure that the statement adheres to our policy

Processed acoustic features and avian community data for all point counts is published at https://doi.org/10.5281/zenodo.7410357. Code with instructions to reproduce analyses, results, and figures can be found at https://zenodo.org/record/7845899.

# Human research participants

Policy information about studies involving human research participants and Sex and Gender in Research.

| | |
|---|---|
| Reporting on sex and gender | N/A |
| Population characteristics | N/A |
| Recruitment | N/A |
| Ethics oversight | N/A |

Note that full information on the approval of the study protocol must also be provided in the manuscript.

# Field-specific reporting

Please select the one below that is the best fit for your research. If you are not sure, read the appropriate sections before making your selection.

☒ Life sciences          ☐ Behavioural & social sciences          ☐ Ecological, evolutionary & environmental sciences

For a reference copy of the document with all sections, see nature.com/documents/nr-reporting-summary-flat.pdf

# Life sciences study design

All studies must disclose on these points even when the disclosure is negative.

| | |
|---|---|
| Sample size | India: Data was collected from 91 sites in the Western Ghats between March 2020 and May 2021. Sites were a mixture of montane wet evergreen and semi-evergreen forests (29), montane grasslands (8), moist deciduous forests (7), timber plantations (17), tea plantations (20), agricultural land (8), and settlements (2). <br> 147 15-minute point counts were conducted between 6-10am at each site (mean 1.6 per site). A variable-distance point count approach was followed, and all bird species heard, seen, and those that flew over (primarily raptor species) were noted. All point counts were carried out between 6am and 10am (timing of high avian activity) at each location. <br> Single channel audio was recorded during each point count using an Audiomoth device6. Recordings were saved in WAV format at a sampling rate of 48 kHz. <br> Malaysia: Data was collected from a varied tropical landscape at the Stability of Altered Forest Ecosystems (SAFE) project24 in Sabah between March 2018 and February 2020. The 14 sites spanned a degradation gradient: two in protected old growth forest, two in a protected riparian reserve, six in selectively logged forest (logging events in 1970s and early 2000s), two in salvage logged forest (last logged in early 2010s), and two in oil palm plantations. Sites had a mean separation distance of 7.6km, with the closest two being 583m apart. <br> 977 20-minute avifaunal point counts were performed across 24 hours of the day at each site (59-80 per site). During point counts all visual or aural encounters of avifaunal species within a 10m radius of the sampling site were recorded. Species identifications and names were validated using the Global Biodiversity Information Facility25 (GBIF). A total of 216 avifaunal species were recorded across all point counts. <br> Single channel audio was recorded during each point count with a Tascam DR-05 recorder mounted to a tripod at chest height (nominal input level -20 dBV, range 20Hz-22kHz). Recordings were saved in WAV format at a sampling rate of 44.1k kHz. <br> Taiwan: Data was collected from 16 tea plantations in the Alishan tea district, located in Chiayi County of Taiwan, between January and November 2022. The tea plantations spanned an elevation gradient from 816-1464 m and were surrounded by secondary broadleaf forests or coniferous plantations. <br> Stereo audio was recorded at a sampling rate of 44.1kHz for one of every fifteen minutes at each site throughout the sampling period. Wildlife Acoustics Song Meter 4 devices were used (mounted to a tree truck or tripod at chest height) and data was saved in WAV format. <br> 176 10-minute avifaunal point counts were conducted in total, on average once per site per month. Every survey was conducted within 3 hours after sunrise. During point counts, the species of every bird individual visually or aurally detected was recorded and its horizontal distance from the observer was estimated as either 0-25m, 25-50m, 50-100m, >100m or flying over.  Our process for matching audio recordings to point counts is provided later in the Methods. <br> USA: Data was collected from a single site in a temperate forest at Sapsucker Woods, Ithaca from January 2016 to December 2021. <br> Audio was recorded continuously using a Gras-41AC precision microphone mounted at chest height and digitised with a Barix Instreamer ADC at a sampling rate of 48 kHz. Files were saved in 15-minute chunks and in WAV format. <br> In the absence of standardised point counts across such a long duration, we used eBird checklists to determine avian communities (only possible since Sapsucker Woods is a major hotspot for eBird data). Data was filtered to only keep checklists which were complete, of the "travelling" or "stationary" types, located within 200m of the recording location, and shorter than 30 minutes. We combined occurrence data from all eBird checklists that started during each audio recording to create one pseudo point count per audio file. Since we only use occurrence data, double counting of the same species was not an issue. This resulted in a total of 6,734 point counts. |
| Data exclusions | N/A |
| Replication | N/A |
| Randomization | N/A |

Blinding                 N/A

# Reporting for specific materials, systems and methods

We require information from authors about some types of materials, experimental systems and methods used in many studies. Here, indicate whether each material, system or method listed is relevant to your study. If you are not sure if a list item applies to your research, read the appropriate section before selecting a response.

### Materials & experimental systems

| n/a | Involved in the study |
|-----|----------------------|
| ☒ ☐ | Antibodies |
| ☒ ☐ | Eukaryotic cell lines |
| ☒ ☐ | Palaeontology and archaeology |
| ☒ ☐ | Animals and other organisms |
| ☒ ☐ | Clinical data |
| ☒ ☐ | Dual use research of concern |

### Methods

| n/a | Involved in the study |
|-----|----------------------|
| ☒ ☐ | ChIP-seq |
| ☒ ☐ | Flow cytometry |
| ☒ ☐ | MRI-based neuroimaging |

