## [Peer Review File · Nature Ecology & Evolution]

Peer Review Information

Journal: Nature Ecology & Evolution

Manuscript Title: Limits to the accurate and generalisable use of soundscapes to monitor biodiversity

Corresponding author name(s): Sarab S. Sethi

Editorial Notes:

Reviewer Comments & Decisions:

Decision Letter, initial version:

28th February 2023

Dear Dr Sethi,

Your manuscript entitled "Is there an accurate and generalisable way to use soundscapes to monitor biodiversity?" has now been seen by two reviewers, whose comments are copied below. The reviewers have raised a number of concerns which we would like to see addressed in a revised manuscript before we can reach a final decision regarding publication in Nature Ecology & Evolution.

We therefore invite you to revise your manuscript taking into account all reviewer and editor comments. Please highlight all changes in the manuscript text file [OPTIONAL: in Microsoft Word format].

* If you have not done so already please begin to revise your manuscript so that it conforms to our Brief Communication format instructions at <http://www.nature.com/natecolevol/info/final-submission>. Refer also to any guidelines provided in this letter.

[REDACTED]

2Note: This URL links to your confidential home page and associated information about manuscripts you may have submitted, or that you are reviewing for us. If you wish to forward this email to co-authors, please delete the link to your homepage.

Nature Ecology & Evolution is committed to improving transparency in authorship. As part of our efforts in this direction, we are now requesting that all authors identified as 'corresponding author' on published papers create and link their Open Researcher and Contributor Identifier (ORCID) with their account on the Manuscript Tracking System (MTS), prior to acceptance. ORCID helps the scientific community achieve unambiguous attribution of all scholarly contributions. You can create and link your ORCID from the home page of the MTS by clicking on 'Modify my Springer Nature account'. For more information please visit www.springernature.com/orcid.

[REDACTED]

Reviewer expertise:

Reviewer #1: Eco-acoustics

Reviewer #2: Soundscape monitoring, machine learning

Reviewers' comments:

Reviewer #1 (Remarks to the Author):

NATECOLEVOL-221218251 Review

Overall comments

This is an important and timely study, succinctly demonstrating the limitations of soundscape methods for estimating biodiversity metrics or ecosystem health. The soundscape field has grown rapidly in

2recent years, yet there generalisability of many proposed metrics, indices and tools between studies is negligible. The quantitative results of this research show the lack of transferability of commonly used soundscape methods as well as more sophisticated between datasets and even within sites for predicting species richness. The authors suggestion that the appropriate use of soundscapes approaches as a measure of community diversity similarity is sensible, whilst also highlighting of the consistent need for caution and ground-truthing. The suggestion of using approach to provide direction for focused passive acoustic studies is also welcome and particularly will help guide conservation practitioners in their use of soundscape approaches. There are only a couple of minor general comments and a number of specific comments that need addressing before publication.

In Figure 1d, it appears that the performance (R²) of the machine learning model trained on the Taiwan dataset and tested on the India dataset was comparable to the within dataset performance of for the USA. Would the authors be able to discuss the reason for this, for instance, could this be due to some similarities in the broader soundscape features (e.g. similar biotic and abiotic sounds) due to geographic features (altitude)?

Would the authors be able to comment on whether there would be merit in testing the transferability of the learned feature method within biomes, or habitat types?

Specific comments

Results

Pg3, 1st para, L3-4: Suggest adjusting to improve sentence flow - "In all four datasets, four learned features and no soundscape indices correlated with avian richness (Fig. 1b)...".

Figure 1a: Please include a label for the brown bars in the legend.

Discussion

Pg5, 1st para, L6: Replace "But, again.." with "However, again,..".

Pg 5, 2nd para L4: Suggest replacing "non-avian species" with "non-avian biotic sounds".

Pg5 3rd para, L5: Please be specific about what type of "interventions" are being referred to here, e.g. "conservation interventions".

Methods

- Avifaunal point counts

o India

1st para, L1: Correct to "Data were collected"

2nd para, L1: It is generally not considered good practice to start a sentence with a numeral, correct to "One hundred and forty-seven", or re-word to e.g. "At each site, 147 15-minute..."

o Malaysia

2nd para, L1: Again, Sentence starts with a numeral, please correct to "Nine hundred and seventy-seven" or re-word, e.g. "At each site, 977 20-minute..."

3rd para, L2: The description of the position of the acoustic recorder as "at chest height" is subjective and not repeatable. Please provide the height in metres.

o Taiwan

32nd para, L2: Possible typo, correct "tree truck" to "tree trunk"?

2nd para, L3: Again, please provide the height of the acoustic sensor in metres.

3rd para, L1: Sentence starts with a numeral, please correct to spell out the number or re-word, as above.

o USA

2nd para, L1: Again, please provide the height of the acoustic sensor in metres.

- Acoustic features

2nd para, L3: Please could the authors explain why the audio was down sampled to 16kHz for the VGGish CNN.

3rd para, L3: Correct to "where audio data were sparser"

- Machine learning predictions of avian richness

Please could the authors explain why for each dataset point counts were split into training and testing sets randomly, rather than by site, which is generally considered to be a more robust method.

Reviewer #2 (Remarks to the Author):

The study investigates whether soundscape analysis provides reliable and generalisable estimates of bird species richness and changes in community composition as an efficient tool for biodiversity monitoring. Using a heterogeneous database composed of soundscape recordings, point counts and bird checklists from four regions, authors test for correlations and predictive performance of machine learning models based on hundreds of acoustic features. The research question is timely as it is under debate to what extent this analytical approach is efficient to assist passive acoustic monitoring in extracting ecologically meaningful information. To address this question, authors opted for a suitable and novel strategy that rely on combining distinct audio datasets, and supported their results using professional and modern statistical analysis. In general, the manuscript is clear and well-written. Nevertheless, I see a series of issues related to the characteristics of the datasets and the computation of the acoustic indices that should be considered and clarified.

Major comments

The four datasets used in the study are highly different in terms of: (i) sampling effort and design (number of sites, number of samples per site, sampling period, etc.), (ii) survey method to estimate bird presence and richness (a variable-distance point count, a 10-m radius point count, a 100-m point count, and non-standardized checklists); and (iii) acoustic equipment to record soundscapes (a distinct device per region). I found this the main weakness of the study as someone can question whether the lack of generalisable estimates might be due to the absence of an harmonized data

4collection rather than an inefficient analytical method. This only influences the assessment of predictive capacity of machine learning models (and not the correlation estimates within each dataset), but I think this is the most novel and relevant result of the study, so this point, in my opinion, matters.

In addition to the methodological differences among regions, bird surveys differ from soundscape recordings in species detectability, as the surveys conducted were based on both aural and visual detection, while audio recordings only captured signals from acoustically active species (not those mostly silent or inactive during the sampling period). This might partially have biased the estimates and reduced the predictive capacity of the soundscape features. I think a more fair and appropriate way to test for this correlation would be to determine species richness directly on the recorded soundscapes (e.g. by aural and visual inspection of spectrograms) so that both methods are based on the same source of information.

With the goal of improving the accuracy of the estimates, I also think that authors might have applied additional adjustments in the computation of the acoustic indices. For instance, it is a common technique (although time-consuming) to characterize sources of noise and remove them using frequency filters or others whenever possible. Besides, some indices allow to focus the analysis in particular frequency bands of the species of interest, excluding other potential sound sources, such as insects or frogs. In this sense, I missed more details related to the computation of the acoustic indices in Methods. Among the 60 indices used, they are highly variable and often require specific settings or parameters to be computed that should be reported in the manuscript, as well as any other pre-processing step conducted before the calculation of the indices.

Another noticeable characteristic of the database is that contains a clearly unbalanced number of observations among regions (from 147 to 6734). First, this might be treated somehow in the analysis as sample size strongly impact on statistical inferences. One would expect to rely more on findings provided by the large dataset and this to be commented in the manuscript. Paradoxically, the large one is also the one with more sources of uncertainty (eBird checklists) and perhaps this has refrained authors to use sample size as an interpretable information. Second, I wonder how authors have deal with unbalanced sampling when using ML models in cross-dataset validation. I would expect models be more robust when trained in thousands of observations than when trained with a hundred. And thus, in my opinion, this should be discussed or explained in the text.

An alternative approach would be to select some of all datasets, excluding those with higher uncertainty. This might be the case of Taiwan dataset, where recordings and bird surveys are not temporally matching in a suitable manner ("audio recordings begun within one-hour window centred around the point count's start time"). It seems to me perhaps a large time lap between measurements.

Finally, I think authors should also clarify a concept they often refer throughout the manuscript (almost in all sections) – "ground-truth calibration data" – but it is likely unclear for most of the readers. The general message is that soundscape approach must only be applied when this kind of data is available (or not even in that case). I guess they mean annotated data to train and test predictive models and to identify the most efficient acoustic features to estimate species richness or

5community change. Thus, the type of data authors used for the study (bird surveys and checklists) and that actually provide evidence that these methods are not fully reliable and generalisable. In Abstract, but mainly in Introduction and Discussion, I think this point needs more precise and coherent explanation.

Minor comments

Abstract

- Please check: "By better understanding how to use interpret data reliably,..."

Introduction

- "...training data, meaning the sonic..."

Training data is a limiting factor constraining the application of ML algorithms because they are costly to collate at a large scale. And this is likely one of the main reasons, together with the strong expertise on programming required for applying ML, why more user-friendly and less time-consuming alternative approaches, such as soundscape analysis, have succeeded in last years. I do not think the need of training (or annotated) data implies that those datasets are taxonomically biased. This mostly depends on the study group and our knowledge on their vocal diversity.

Results

- "First, we investigated univariate..."

I would add "within each dataset" or "for region independently" at the end of the sentence to clarify the approach (although this is also explained later in Methods).

- "... many LFs and SSIs correlated with..."

I suggest to add "significantly" to the sentence

- "... number varied greatly by dataset (Fig. 1a). Four learned... "

Consider to rephrase, e.g. "... greatly depending on the dataset (Fig. 1a). However, only four learned..."

- "... whilst most features..."

I suggest to use "soundscape features" or any other term to ensure the reader understand that they include both "soundscape indices and learned features", and not only "learned features".

- "105 x LFs, 43 x SSIs"

It is unclear to me what "x" mean here. 105 LFs? If so, I would remove "x" (as in the caption of Fig. 1).

- "However, the models were unable to predict..."

Please also indicate R2 for these models

- Could you add the scale in y-axis of the panel A?
- Could you also add sample size (N) in each sub-panel of panel A, C, and E, or in the caption?
- I am not sure if I am understanding the panel B correctly. Have you compared the features correlated with richness among each datasets and counted the ones that are the same when comparing 2, 3 or 4 datasets? If so, there are several possible combinations of dataset to be compared and then I would expect to see a mean and bar error in the figure (rather than a single bar). Moreover, why there is zero in x-axis (number of datasets)? Why is necessary the vertical dashed line (number of datasets)? I guess I am likely missing something... Then, I might understand wrongly the related sentence "Four learned features and no soundscape indices correlated with avian richness in all four datasets (Fig. 1b)". Alternatively, have you combined datasets in a single one and compute new correlations? I do not find this explained in Methods.

Discussion

- Please consider rephrase and/or expand the last paragraph, and particularly the third sentence, which is unclear to me. I think readers would appreciate more concrete and clear proposals as alternative approaches to improve the use of soundscapes analysis in biodiversity monitoring. I found these lines too vague or confusing.

Methods

- For the sake of consistency, I suggest the first section "Avifaunal point counts" to include the same information and following the same order for each region.
- India. The last sentence of the second paragraph provides repeated information. Please add "separation distance" between sites (at the end of the second sentence) and "number of species counted" (at the end of the second paragraph), as reported for Malaysia.
- Malaysia. Indicate that Tascam recorder includes omnidirectional mics.
- Taiwan and USA. I would change the position of the second and third paragraphs to follow the same order than in the previous regions.
- Indicate the polar pattern of the Gras microphone
- "... and shorter than 30 minutes"
I do not understand what you mean
- "...fitting a 1st order polynomial..."
Here I would indicate that you selected only two features and the rationale of that choice.
- Are the Euclidean distance calculated within a 188-dimension space?

- I am wondering why ML models were fitted with only one type of features (LFs or SSIs), instead of both together. Would not this potentially improve their predictive performance?

*****END*****

Author Rebuttal to Initial commentsResponse to reviewers: Is there an accurate and generalisable way to use soundscapes to monitor biodiversity?

NATECOLEVOL-221218251

Reviewer #1

This is an important and timely study, succinctly demonstrating the limitations of soundscape methods for estimating biodiversity metrics or ecosystem health. The soundscape field has grown rapidly in recent years, yet the generalisability of many proposed metrics, indices and tools between studies is negligible. The quantitative results of this research show the lack of transferability of commonly used soundscape methods as well as more sophisticated between datasets and even within sites for predicting species richness. The authors suggestion that the appropriate use of soundscapes approaches as a measure of community diversity similarity is sensible, whilst also highlighting of the consistent need for caution and ground-truthing. The suggestion of using approach to provide direction for focused passive acoustic studies is also welcome and particularly will help guide conservation practitioners in their use of soundscape approaches. There are only a couple of minor general comments and a number of specific comments that need addressing before publication.

Thank you for your kind and considered comments. We hope that we have now addressed the minor issues you raised in your review.

In Figure 1d, it appears that the performance (R^2) of the machine learning model trained on the Taiwan dataset and tested on the India dataset was comparable to the within dataset performance of for the USA. Would the authors be able to discuss the reason for this, for instance, could this be due to some similarities in the broader soundscape features (e.g. similar biotic and abiotic sounds) due to geographic features (altitude)?

This certainly was interesting, and it could be caused by the two datasets sharing similar axes of soundscape change when species richness was increased. However, the fact that there was not any relationship between predicted and true species richness in the converse case, i.e., when models were trained on India and tested on Taiwan ($R^2 < 0$), suggests that this result could just have been down to chance. A full detailed investigation of which features in particular shared correlations across the two datasets may start to unpick this question, but we believe it would lie outside the scope of our study.

Additionally, it's important to note that the relationship between true richness and predicted richness when the model was trained on Taiwan and tested on India was still very low ($R^2 = 0.13$). We hope the slightly rephrased final paragraph in the discussion (L144) makes this point even more clearly now.

Would the authors be able to comment on whether there would be merit in testing the transferability of the learned feature method within biomes, or habitat types?

Whilst beyond the scope of the questions we consider in our study, this would certainly be interesting to investigate. One would probably require multiple independent datasets from closely matched biomes to begin to address this, which unfortunately we didn't have access to. However, we have included a sentence in the Discussion to note this may be an interesting future path of research.

9

ess
: is

L130: The diversity of flora, fauna, survey designs, and recording equipment across the datasets might, perhaps, make these results unsurprising. Indeed, further studies may show limited generalisability can be achieved when study biomes and survey methodologies are closely matched.

Specific comments

Results

Pg3, 1st para, L3-4: Suggest adjusting to improve sentence flow - "In all four datasets, four learned features and no soundscape indices correlated with avian richness (Fig. 1b)...".

Rephrased integrating similar comments from Reviewer 2.

L79: Most acoustic features correlated significantly with species richness in either one or two of the four datasets (105 LFs, 43 SSIs; Fig. 1b). Only four LFs and not a single SSI correlated with avian richness across all four datasets (Fig. 1b).

Figure 1a: Please include a label for the brown bars in the legend.

The bars that appear brown are simply where the orange (SSI) bars overlap with the blue (LF) bars. This has been clarified now in the figure caption.

L96: ... and overlaps between the SSI (orange) and LF (blue) histograms are in brown.

Discussion

Pg5, 1st para, L6: Replace "But, again.." with "However, again,..".

Corrected.

Pg 5, 2nd para L4: Suggest replacing "non-avian species" with "non-avian biotic sounds".

Corrected.

Pg5 3rd para, L5: Please be specific about what type of "interventions" are being referred to here, e.g. "conservation interventions".

Corrected.

Methods: Avifaunal point counts

India: 1st para, L1: Correct to "Data were collected"

10

We have throughout referred to data in the singular – as is accepted by the Merriam-Webster dictionary: <https://www.merriam-webster.com/dictionary/data>. Our preference is to leave it as is (but we are also not going to die on this hill!)

ess
: is

India: 2nd para, L1: It is generally not considered good practice to start a sentence with a numeral, correct to “One hundred and forty-seven”, or re-word to e.g. “At each site, 147 15-minute...”

Malaysia: 2nd para, L1: Again, Sentence starts with a numeral, please correct to “Nine hundred and seventy-seven” or re-word, e.g. “At each site, 977 20-minute...”

Taiwan: 3rd para, L1: Sentence starts with a numeral, please correct to spell out the number or re-word, as above.

We have rephrased these sentences so they no longer start with numerals.

L240: *In total, 147 15-minute point counts were conducted between 6-10am (mean 1.6 per site).*

L251: *In total, 977 20-minute avifaunal point counts were performed across 24 hours of the day (59-80 per site).*

L261: *In total, 176 10-minute avifaunal point counts were conducted, on average once per site per month.*

Malaysia: 3rd para, L2: The description of the position of the acoustic recorder as “at chest height” is subjective and not repeatable. Please provide the height in metres.

Taiwan: 2nd para, L3: Again, please provide the height of the acoustic sensor in metres.

USA: 2nd para, L1: Again, please provide the height of the acoustic sensor in metres.

In India, Taiwan, and Malaysia the recorder was placed at approximately chest height by field technicians – we have now however replaced “chest height” with “1-2m raised from the ground” to provide more quantitative estimates. In USA the single microphone was raised approximately 1.5m from the ground. Recorded audio does not vary significantly on such small spatial scales (1m), and therefore we believe this small variability will not influence the quality of our data.

L244: *... Audiomoth device raised 1-2m from the ground.*

L255: *... mounted to a tripod raised 1-2m from the ground...*

L268: *... (mounted to a tree trunk or tripod raised 1-2m from the ground) ...*

L280: *... precision microphone raised approximately 1.5m from the ground...*

Taiwan: 2nd para, L2: Possible typo, correct “tree truck” to “tree trunk”?

Yes, just a typo. Corrected now.

Methods: Acoustic features

11

2nd para, L3: Please could the authors explain why the audio was down sampled to 16kHz for the VGGish CNN.

ess
: is

When using any CNN model, all input data must be transformed into the same format as was used when training the model. Since we used a pre-trained model, VGGish, we had to follow the same data transformations used by Hershey et al., 2017, which required audio to be resampled to 16kHz. This has been clarified in the text now:

L295: *First, to transform raw audio into the input format expected by the pre-trained VGGish model...*

3rd para, L3: Correct to “where audio data were sparser”

See above comment regarding data as singular vs plural.

Methods: Machine learning predictions of avian richness

Please could the authors explain why for each dataset point counts were split into training and testing sets randomly, rather than by site, which is generally considered to be a more robust method.

Train/test splits were chosen randomly due to the varying nature of our four datasets. Since USA only had one site and many replicates (total 6,734 point counts), in India we had 91 sites with few replicates (total 119 point counts, mean 1.6 per site). This meant splitting point counts randomly was the only systematic approach we could apply in the same way to all of our datasets, without potentially introducing biases in our selection.

Nevertheless, we do certainly appreciate that splitting by site is commonly done in the literature, and perhaps our method of randomly splitting samples represents a “best-case” scenario. One would expect non-random splits of training and test data (e.g., by site, as suggested) would lead to even further reduced accuracy of species richness predictions and further reduced generalisability of analyses as train and test data characteristics would diverge further. This would strengthen our core results, indicating that there are no common soundscape markers of biodiversity. We have now added a short justification of our method of splitting into our Methods.

L323: *Splitting train and test data randomly represents a “best-case” scenario since, within datasets, training data distributions will closely match test data distributions.*

Reviewer #2

The study investigates whether soundscape analysis provides reliable and generalisable estimates of bird species richness and changes in community composition as an efficient tool for biodiversity monitoring. Using a heterogenous database composed of soundscape recordings, point counts and bird checklists from four regions, authors test for correlations and predictive performance of machine learning models based on hundreds of acoustic features. The research question is timely as it is under debate to what extent this analytical approach is efficient to assist passive acoustic monitoring in extracting ecologically meaningful information. To address this question, authors opted for a suitable and novel strategy that rely on combining distinct audio datasets, and supported their results using professional and modern statistical analysis. In general, the manuscript is clear and well-written. Nevertheless, I see a series of issues related to the characteristics of the datasets and the computation of the acoustic indices that should be considered and clarified.

12

ess
: is

Thank you for your kind and considered comments. We hope that in this revision we have now sufficiently addressed both concerns regarding the dataset characteristics and computation of acoustic indices as well as the minor comments.

Major comments

The four datasets used in the study are highly different in terms of: (i) sampling effort and design (number of sites, number of samples per site, sampling period, etc.), (ii) survey method to estimate bird presence and richness (a variable-distance point count, a 10-m radius point count, a 100-m point count, and non-standardized checklists); and (iii) acoustic equipment to record soundscapes (a distinct device per region). I found this the main weakness of the study as someone can question whether the lack of generalisable estimates might be due to the absence of an harmonized data collection rather than an inefficient analytical method. This only influences the assessment of predictive capacity of machine learning models (and not the correlation estimates within each dataset), but I think this is the most novel and relevant result of the study, so this point, in my opinion, matters.

In response to points (i), (ii), and (iii), the variation in survey designs, species survey methodologies, and audio recorders may indeed have an effect on the generalisability results. However, throughout the study we applied very conservative data processing to account for this variability. For example, we only consider species occurrence data since we explicitly acknowledged that the quality of abundance data differed more severely across datasets (L284). Additionally, all recorders were deployed at approximately breast height, and downsampled to a common sampling frequency before calculating acoustic features to follow best practices and minimise biases in the audio.

However, the species community data and acoustic recordings collected in this study spanned multiple nations, years, and projects, meaning that controlling for every parameter was not feasible. Furthermore, collecting fresh data at this scale would be impossible. We have, nevertheless, now explicitly acknowledged this point in the first paragraph of our discussion. Furthermore, we have acknowledged that our recommendation is in the context of “current state-of-the-art” and that perhaps additional work may achieve generalisability in limited scenarios.

L130: The diversity of flora, fauna, survey designs, and recording equipment across the datasets might, perhaps, make these results unsurprising. Indeed, further studies may show limited generalisability can be achieved when study biomes and survey methodologies are closely matched. Nevertheless, given current state-of-the-art, our results stress that however well an acoustic feature or ML model may perform in one scenario, without access to high quality ground truth data from the exact location being studied, soundscape methods should not be used to generate predictions of species richness.

However, our point made in the third paragraph of the discussion relating to the relatively poor accuracy of soundscape methods – even within datasets – still stands. We have elaborated on this point to make it clearer that even if large, standardised soundscape datasets can be collected, that it may not be a golden bullet.

L145: This result suggests that even if we can collate standardised global soundscape datasets, and train models using ground truth data from every biome on earth, we may still be left wanting.

13

ess
: is

In addition to the methodological differences among regions, bird surveys differ from soundscape recordings in species detectability, as the surveys conducted were based on both aural and visual detection, while audio recordings only captured signals from acoustically active species (not those mostly silent or inactive during the sampling period). This might partially have biased the estimates and reduced the predictive capacity of the soundscape features. I think a more fair and appropriate way to test for this correlation would be to determine species richness directly on the recorded soundscapes (e.g. by aural and visual inspection of spectrograms) so that both methods are based on the same source of information.

It should first be noted that all point count surveys were audio-visual in nature (as detailed in the Methods already), and as such one would expect any data derived from manual listening of the spectrograms to simply be a subset of the avian occurrence data that we already have.

Whilst the exact details of the bird surveys differ from one location to another, the common thread between all of them is that they were conducted by a trained expert (or experienced citizen scientist through eBird) who physically visited the sites of interest and surveyed biodiversity. Whilst any data source is imperfect, data of the type we used for calibration would not normally be questioned for its quality when presented in an ecological study. However, since our study aimed to ask questions around the reliability and trustworthiness of soundscape derived measures of biodiversity, we needed a "gold-standard" to aspire to.

Deriving ground-truth data for a subset of our recordings would certainly be possible (but time-consuming) by manual listening. However, this would present a cyclical situation where features of the audio are calibrated using only data from the audio files themselves. When using acoustic monitoring in practice we are usually concerned with how we can track the full spectrum of biodiversity, rather than only the species which are vocalising. In this respect, we believe using ground-truth data which was derived from the audio files themselves (and would necessarily miss species which were present and noted by in-person surveyors, but not vocalising) would in fact be less desirable.

With the goal of improving the accuracy of the estimates, I also think that authors might have applied additional adjustments in the computation of the acoustic indices. For instance, it is a common technique (although time-consuming) to characterize sources of noise and remove them using frequency filters or others whenever possible. Besides, some indices allow to focus the analysis in particular frequency bands of the species of interest, excluding other potential sound sources, such as insects or frogs. In this sense, I missed more details related to the computation of the acoustic indices in Methods. Among the 60 indices used, they are highly variable and often require specific settings or parameters to be computed that should be reported in the manuscript, as well as any other pre-processing step conducted before the calculation of the indices.

Many of the soundscape indices we used already focus on only specific frequency bands in their analysis. For example; BI (the Bioacoustics Index) only considers areas in the spectrogram between 2-15kHz; NDSI assumes anthropophony is between 0-1kHz and biophony is between 1-10kHz, etc. Tuning these parameters to each dataset would theoretically be possible, but to ascertain exactly how to tune these parameters to each dataset would be a vast amount of work – as already noted by the reviewer. Furthermore, it would be difficult to systematically tune these to each dataset even if we were to try, and this could introduce new sources of bias into our results.

As requested, we have added a full list of the 60 soundscape indices (SSIs) used in our study in a new supplementary table (SI Table S5). Each SSI can have multiple tuneable parameters, and since we did not modify any from their default values, listing them all would be overly cumbersome and unhelpful.

Table S5: List of all 60 scikit-maad v1.3 features used and the functions used to generate them – referred to as soundscape indices (SSIs) throughout the study. In all cases, default parameters were used, details of which can be found within the official documentation <https://scikit-maad.github.io/>.

Feature	scikit-maad v1.3 function	Feature	scikit-maad v1.3 function
ZCR	all_temporal_alpha_indices	EPS_KURT	all_spectral_alpha_indices
MEANf	all_temporal_alpha_indices	EPS_SKEW	all_spectral_alpha_indices
VARf	all_temporal_alpha_indices	ACI	all_spectral_alpha_indices
SKWEWf	all_temporal_alpha_indices	NDSI	all_spectral_alpha_indices
KURTI	all_temporal_alpha_indices	rBA	all_spectral_alpha_indices
LEQf	all_temporal_alpha_indices	AnthroEnergy	all_spectral_alpha_indices
BGNf	all_temporal_alpha_indices	BioEnergy	all_spectral_alpha_indices
SNRf	all_temporal_alpha_indices	BI	all_spectral_alpha_indices
MED	all_temporal_alpha_indices	ROU	all_spectral_alpha_indices
Hf	all_temporal_alpha_indices	ADI	all_spectral_alpha_indices
ACTIFraction	all_temporal_alpha_indices	AEI	all_spectral_alpha_indices
ACTICount	all_temporal_alpha_indices	LFC	all_spectral_alpha_indices
ACTIMean	all_temporal_alpha_indices	MFC	all_spectral_alpha_indices
EVNI:Fraction	all_temporal_alpha_indices	HFC	all_spectral_alpha_indices
EVNI:Mean	all_temporal_alpha_indices	ACTspFract	all_spectral_alpha_indices
EVNI:Count	all_temporal_alpha_indices	ACTspCount	all_spectral_alpha_indices
MEANf	all_spectral_alpha_indices	ACTspMean	all_spectral_alpha_indices
VARf	all_spectral_alpha_indices	EVNspFract	all_spectral_alpha_indices
SKWEWf	all_spectral_alpha_indices	EVNspMean	all_spectral_alpha_indices
KURTI	all_spectral_alpha_indices	EVNspCount	all_spectral_alpha_indices
NBPEAKS	all_spectral_alpha_indices	TFSD	all_spectral_alpha_indices
LEQf	all_spectral_alpha_indices	H_Havrd	all_spectral_alpha_indices
ENRf	all_spectral_alpha_indices	H_Renyi	all_spectral_alpha_indices
BGNf	all_spectral_alpha_indices	H_pairedShannon	all_spectral_alpha_indices
SNRf	all_spectral_alpha_indices	H_gamma	all_spectral_alpha_indices
Hf	all_spectral_alpha_indices	H_GiniSimpson	all_spectral_alpha_indices
EAS	all_spectral_alpha_indices	RAOQ	all_spectral_alpha_indices
ECU	all_spectral_alpha_indices	AGI	all_spectral_alpha_indices
ECV	all_spectral_alpha_indices	ROItotal	all_spectral_alpha_indices
EPS	all_spectral_alpha_indices	ROIcover	all_spectral_alpha_indices

In addition to the supplementary table, we have also added references to the exact version of the scikit-maad library that we are using to aid in reproducibility, as well as a brief justification for using the default parameters when generating features.

L287: We used two approaches to calculate acoustic features for each recording. The first was using a suite of 60 soundscape indices (SSIs), using the scikit-maad library (v1.3)²¹. Specifically, we appended together the feature vectors from the functions `maad.features.all_temporal_alpha_indices` and `maad.features.all_spectral_alpha_indices`. In all cases we used default arguments for generating features, since tuning parameters to each of the datasets would have been a cumbersome and inherently qualitative process susceptible to introducing bias. A full list of features is given in SI Table S5.

Another noticeable characteristic of the database is that contains a clearly unbalanced number of observations among regions (from 147 to 6734). First, this might be treated somehow in the analysis as sample size strongly impact on statistical inferences. One would expect to rely more on findings provided by the large dataset and this to be commented in the manuscript. Paradoxically, the large one is also the one with more sources of uncertainty (eBird checklists) and perhaps this has refrained authors to use sample size as an interpretable information. Second, I wonder how authors have deal with unbalanced sampling when using ML models in cross-dataset validation. I would expect models

be more robust when trained in thousands of observations than when trained with a hundred. And thus, in my opinion, this should be discussed or explained in the text.

This was our intuition too, but in fact it didn't turn out to be true. To make this point more quantitatively, we have added a new analysis to the manuscript investigating the correlation between R^2 values in Fig. 1d and dataset sample sizes.

We found no correlation between within-dataset or mean cross-dataset R^2 values and dataset sample sizes.

L88: *There was no significant correlation between dataset sample sizes and within-dataset or mean cross-dataset R^2 values (Pearson's correlation, $p > 0.05$).*

We have also added details on this analysis to the relevant part of the Methods:

L328: *We used Pearson's correlation coefficient to test whether there was a link between dataset sample sizes and classifier performance. For within dataset performance, we used the diagonal values of Fig. 1d. For cross-dataset performance, we took the mean of each row in Fig. 1d, excluding the diagonal values.*

The reasons for the lack of correlation could be related to varying sources of uncertainty, but they could also be due to varying complexity of soundscapes between the different biomes, varying spatial/temporal resolutions of the datasets, or many other factors. Investigating exactly which dataset characteristics are most important for soundscape prediction accuracy is certainly an interesting question. However, it is a question that is not answerable using the small number of datasets we have within this study.

An alternative approach would be to select some of all datasets, excluding those with higher uncertainty. This might be the case of Taiwan dataset, where recordings and bird surveys are not temporally matching in a suitable manner ("audio recordings begun within one-hour window centred around the point count's start time"). It seems to me perhaps a large time lag between measurements.

We added a new analysis and supplementary figure to evaluate the sensitivity of our results to the size of the window used in the Taiwan analysis.

Instead of the one-hour window (which is still used for results in the main text) we re-matched audio recordings to point counts using 20-minute and 40-minute window sizes instead. We found that our core results were unaffected by this choice in data processing parameter.

We have added the below sentence in the methods and have added SI Fig S3 to reproduce Fig 1d but with the alternative window sizes.

L305: *Using window sizes of 40 minutes or 20 minutes around the point count start time in Taiwan did not significantly change results (Fig 1d reproduced with different window sizes, SI Fig. S4).*

Figure S4: The chosen size of the window around the point count start time in the Taiwan dataset did not change the accuracy or generalisability of the machine learning models trained to predict species richness.

Most of the panels in Figure 1 already allow interpretation for how things would pan out with a subset of the datasets. For example, Fig. 1b shows that still a very small number of features (SSIs or LFs) correlate with 3 of the 4 datasets – and most correlate with only 0 or 1 dataset. Fig. 1d would look almost identical if we were to reduce the number of datasets (except for deleting the dataset’s respective row/column). I.e., the issue we are seeing is not that features/predictive models are not generalising across **all** datasets but rather they are hardly generalising across **any** of the datasets we are looking at.

Additionally, we believe the variability within the datasets we have presented is representative of the broader soundscape monitoring literature – a point we hope we have already addressed in our response to the first major comment.

Finally, I think authors should also clarify a concept they often refer throughout the manuscript (almost in all sections) – “ground-truth calibration data” – but it is likely unclear for most of the readers. The general message is that soundscape approach must only be applied when this kind of data is available (or not even in that case). I guess they mean annotated data to train and test predictive models and to identify the most efficient acoustic features to estimate species richness or community change. Thus, the type of data authors used for the study (bird surveys and checklists) and that actually provide evidence that these methods are not fully reliable and generalisable. In Abstract, but mainly in Introduction and Discussion, I think this point needs more precise and coherent explanation.

We apologise for the slightly cryptic use of “ground-truth data”. Through our answer to your second major comment, we hope we have clarified why we chose to use the data we did for ground-truth calibration.

We have now also clarified our terminology when referring to ground truth data throughout the manuscript (including the abstract, introduction, and discussion sections).

Abstract L26: *This limits the insight that can be derived from audio recordings in regions without independently collected, trusted sources of ground-truth biodiversity data.*

Abstract **L36**: *Therefore, soundscape monitoring should only be used when high quality ground-truth biodiversity data already exists or can be collected for calibration purposes, and in conjunction with more targeted and accurate in-person ecological surveys.*

Intro **L56**: *... when calibrated using trusted independent sources of ground-truth biodiversity data (hereafter, ground-truth data)^{14,15}.*

Discussion **L125**: *Without access to independent sources of trusted ground-truth biodiversity data,...*

Discussion **L144**: *Despite having access to vast amounts of manually collected avifaunal point count data,...*

Minor comments

Abstract

- Please check: "By better understanding how to use interpret data reliably,..."

Apologies, this was a typo. Corrected.

L38: *By better understanding how to interpret data reliably, we hope to broaden the scale...*

Introduction

- "...training data, meaning the sonic..."

Training data is a limiting factor constraining the application of ML algorithms because they are costly to collate at a large scale. And this is likely one of the main reasons, together with the strong expertise on programming required for applying ML, why more user-friendly and less time-consuming alternative approaches, such as soundscape analysis, have succeeded in last years. I do not think the need of training (or annotated) data implies that those datasets are taxonomically biased. This mostly depends on the study group and our knowledge on their vocal diversity.

What we meant was that due to the difficulty of curating large training datasets, creating reliable classifiers for rare species is difficult. Therefore, often the performances of vocalisation detection algorithms are consequently biased towards common species. We have added a citation now to support this claim, and rephrased to clarify that the algorithms themselves are not necessarily biased, just our ability to create reliable algorithms is the issue.

L52: However, vocalisation detection algorithms rely on vast amounts of training data¹¹, meaning even state-of-the-art models are only able to reliably detect vocalisations from the most commonly found species¹²

Results

- "First, we investigated univariate..."

I would add "within each dataset" or "for region independently" at the end of the sentence to clarify the approach (although this is also explained later in Methods).

Added "within each dataset" (**L76**).

18

ess
: is

- "... many LFs and SSIs correlated with..."

I suggest to add "significantly" to the sentence

Added "significantly" (L77).

- "... number varied greatly by dataset (Fig. 1a). Four learned..."

Consider to rephrase, e.g. "... greatly depending on the dataset (Fig. 1a). However, only four learned..."

Rephrased integrating similar comments from Reviewer 1.

L79: *Most acoustic features correlated significantly with species richness in either one or two of the four datasets (105 LFs, 43 SSIs; Fig. 1b). Only four LFs and not a single SSI correlated with avian richness across all four datasets (Fig. 1b).*

- "... whilst most features..."

I suggest to use "soundscape features" or any other term to ensure the reader understand that they include both "soundscape indices and learned features", and not only "learned features".

We would rather avoid using "soundscape features" as this might easily be confused with soundscape indices. We have rephrased this to "acoustic features" instead.

L79: *Most acoustic features correlated significantly ...*

- "105 x LFs, 43 x SSIs"

It is unclear to me what "x" mean here. 105 LFs? If so, I would remove "x" (as in the caption of Fig. 1).

Removed "x" as requested.

- "However, the models were unable to predict..."

Please also indicate R2 for these models

This sentence is referring to all cases where the training and test datasets were not the same, so listing all 12 R² values would be overly long. We have, however, now rephrased this part to include more details on the R² values (also integrating similar comments from Reviewer 1).

L86: *However, except in one case (trained on Taiwan, tested on India, R² = 0.13), the models were unable to predict richness when evaluated on datasets that they were not trained upon (R² < 0, Fig. 1d).*

19

- Could you add the scale in y-axis of the panel A?

Added scale to y-axis (and slightly adjusted so all sub-panels of A share the same y-axis).

ess
: is

- Could you also add sample size (N) in each sub-panel of panel A, C, and E, or in the caption?

We have added sample sizes to the end of the figure caption:

L106: *Number of samples per dataset: India (N=147), Malaysia (N=977), Taiwan (N=165), and USA (N=6,734).*

- I am not sure if I am understanding the panel B correctly. Have you compared the features correlated with richness among each datasets and counted the ones that are the same when comparing 2, 3 or 4 datasets? If so, there are several possible combinations of dataset to be compared and then I would expect to see a mean and bar error in the figure (rather than a single bar). Moreover, why there is zero in x-axis (number of datasets)? Why is necessary the vertical dashed line (number of datasets)? I guess I am likely missing something... Then, I might understand wrongly the related sentence "Four learned features and no soundscape indices correlated with avian richness in all four datasets (Fig. 1b)". Alternatively, have you combined datasets in a single one and compute new correlations? I do not find this explained in Methods.

For each feature, we looked at each dataset in turn and noted whether it significantly correlated with species richness or not. For example, if feature X correlated significantly with species richness in India and Taiwan datasets, but there was no significant correlation in the USA or Malaysia datasets, it would contribute to the bars labelled "2". Likewise, if feature Y correlated with species richness in Taiwan and USA, but not Malaysia or India, it would also be added to the "2" bar in the histogram.

This process was repeated for all 128 LFs and 60 SSIs to build up the histogram shown – and the features are separated by their groups (blue bars for LFs and orange for SSIs). Since it is a deterministic process, there is no way to add uncertainties (through error bars) to the histogram. The "0" column in the x axis represents all of the features that did not correlate with species richness significantly in any of the four datasets.

The dashed line at "4" is not strictly necessary, but helps to visualise the "optimal" state – i.e., all features in the "4" column correlated with species richness significantly in all datasets. However, there were only 4 LFs which correlated significantly with species richness across all datasets (one of them, LF 14, is visualised in panel c) and no SSIs correlated significantly with species richness in all of the datasets.

We have reworded how we present these results to clarify the analyses we performed and their implications:

L79: *Most acoustic features correlated significantly with species richness in either one or two of the four datasets (105 LFs, 43 SSIs; Fig. 1b). Only four LFs and not a single SSI correlated with avian richness across all four datasets (Fig. 1b).*

Discussion

- Please consider rephrase and/or expand the last paragraph, and particularly the third sentence, which is unclear to me. I think readers would appreciate more concrete and clear proposals as alternative approaches to improve the use of soundscapes analysis in biodiversity monitoring. I found these lines too vague or confusing.

20

ess
: is

We have added an extra sentence to fully detail one example of how acoustics could direct detailed monitoring efforts.

L149: *For example, data from large-scale acoustic monitoring networks could be used to direct in-person surveys towards only the sites with large unexpected soundscape changes or shifts in predicted biodiversity.*

Methods

- For the sake of consistency, I suggest the first section “Avifaunal point counts” to include the same information and following the same order for each region.

- Taiwan and USA. I would change the position of the second and third paragraphs to follow the same order than in the previous regions.

We have reordered these sections of the methods (along with some small wording changes) to follow the same structure for consistency.

- India. The last sentence of the second paragraph provides repeated information.

Apologies – the repetition of the hours of the day has now been removed.

L240: *In total, 147 15-minute point counts were conducted (mean 1.6 per site). All point counts were conducted between 6 and 10am as these were the hours with highest expected avian activity.*

- India: Please add “separation distance” between sites (at the end of the second sentence) and “number of species counted” (at the end of the second paragraph), as reported for Malaysia.

We have now added number of species counted to the India, Taiwan, and USA sections of the methods (**L243, L265, L278**).

We have also now added mean and minimum site separation distance for India.

L239: *Sites had a mean separation distance of 18.9km, with the closest two being 824m apart.*

- Malaysia. Indicate that Tascam recorder includes omnidirectional mics.

- USA. Indicate the polar pattern of the Gras microphone

Added directionality of both microphones.

L256: *... (integrated omnidirectional microphone, nominal input level -20 dBV, range 20Hz-22kHz).*

L279: *...using an omnidirectional Gras-41AC precision microphone...*

- “... and shorter than 30 minutes”

I do not understand what you mean

eBird checklists are of varying durations, so we filtered for only checklists that were shorter than 30 minutes to ensure our data wasn't biased by a few outliers which lasted many hours. Rephrased to clarify this:

L275: *...and lasted less than 30 minutes (to minimise bias from varying sampling intensities).*

- "...fitting a 1st order polynomial..."

Here I would indicate that you selected only two features and the rationale of that choice.

In this section, titled "Univariate correlations...", we are only looking at univariate relationships – i.e., how each of the 60 SSIs and 128 LFs independently correlate with species richness. Therefore we are considering one feature at a time – not two as suggested by the reviewer. We have reworded this sentence to clarify:

L316: *Lines of best fit in Fig. 1c were determined by fitting a 1st order polynomial to the data on avian richness (x-axis) and the single acoustic feature being considered (y-axis).*

- Are the Euclidean distance calculated within a 188-dimension space?

Distances were computed in 128D space for the LFs (Fig. 1e) and 60D space for the SSIs (Fig. S3). This has now been clarified in the text.

L335: *Distances were computed in the full 128-dimensional space for the LFs (Fig. 1e), and 60-dimensional space for the SSIs (Fig. S3).*

- I am wondering why ML models were fitted with only one type of features (LFs or SSIs), instead of both together. Would not this potentially improve their predictive performance?

Following this comment, we have run a new analysis investigating whether combining all LFs and SSIs together makes any difference to predictive power and/or generalisability of models.

We found that similar patterns were seen when training ML models using all features together. We have accordingly added a subpanel to SI Fig S1 and a sentence referring to this figure in the results.

L89: *Similar results were found when using only the 60 soundscape indices or all 188 features (128 LFs + 60 SSIs) together to predict richness (SI Fig. S1).*

We have also added details of this analysis to the relevant section of the Methods.

L320: *In SI Fig. S1b, the mean of all 60 SSIs and 128 LFs were combined to create one 188-dimensional feature vector per audio recording.*

New version of SI Fig. S1:

Figure S1: When training and test datasets were the same, predictions of species richness from a machine learning model (Random Forest Regressor) trained on (a) the 60 soundscape indices (SSIs) or (b) all 188 acoustic features (60 SSIs + 128 LFs) were of varying levels of accuracy ($R^2=0.09, 0.50, 0.24, 0.10$ [SSIs] $R^2=0.30, 0.54, 0.24, 0.15$ [all 188 features] for India, Malaysia, Taiwan, USA, respectively). In all cases, predictive models did not generalise to produce informative estimates of species richness when applied to datasets they were not trained on. Similar results using only the 128 LFs are shown in Fig. 1c in the main text.

Decision Letter, first revision:

13th June 2023

Dear Dr. Sethi,

Thank you for submitting your revised manuscript "Is there an accurate and generalisable way to use soundscapes to monitor biodiversity?" (NATECOLEVOL-221218251A). It has now been seen again by the original reviewers and their comments are below. The reviewers find that the paper has improved in revision, and therefore we'll be happy in principle to publish it in Nature Ecology & Evolution, pending minor revisions to satisfy the reviewers' final requests and to comply with our editorial and formatting guidelines.

[REDACTED]

Reviewer #1 (Remarks to the Author):

The authors have addressed all of my comments and given valid responses and I do not have anything further. Based on the responses to mine and the other reviewer's comments I would be happy to see this article published.

Reviewer #2 (Remarks to the Author):

I appreciate that the authors have worked hard to revise and improve the manuscript according to the reviewers' comments. All the main concerns and suggestions that I raised have been successfully addressed, implemented or clarified. This reviewer thanks the authors for their detailed replies to those points and for adjusting the manuscript following most of the suggestions. The additional analyses and extra supplementary material included in this revised version also reinforce some key points and enhance the quality of the paper. Overall, a neat revision of an already good manuscript

24that I consider it is ready to be published. Congratulations for an excellent work and a valuable contribution to the field.

Our ref: NATECOLEVOL-221218251A

23rd June 2023

Dear Dr. Sethi,

Thank you for your patience as we've prepared the guidelines for final submission of your Nature Ecology & Evolution manuscript, "Is there an accurate and generalisable way to use soundscapes to monitor biodiversity?" (NATECOLEVOL-221218251A). Please carefully follow the step-by-step instructions provided in the attached file, and add a response in each row of the table to indicate the changes that you have made. Please also check and comment on any additional marked-up edits we have proposed within the text. Ensuring that each point is addressed will help to ensure that your revised manuscript can be swiftly handed over to our production team.

****We would like to start working on your revised paper, with all of the requested files and forms, as soon as possible (preferably within two weeks). Please get in contact with us immediately if you anticipate it taking more than two weeks to submit these revised files.****

In recognition of the time and expertise our reviewers provide to Nature Ecology & Evolution's editorial process, we would like to formally acknowledge their contribution to the external peer review of your manuscript entitled "Is there an accurate and generalisable way to use soundscapes to monitor biodiversity?". For those reviewers who give their assent, we will be publishing their names alongside the published article.

25Nature Ecology & Evolution offers a Transparent Peer Review option for new original research manuscripts submitted after December 1st, 2019. As part of this initiative, we encourage our authors to support increased transparency into the peer review process by agreeing to have the reviewer comments, author rebuttal letters, and editorial decision letters published as a Supplementary item. When you submit your final files please clearly state in your cover letter whether or not you would like to participate in this initiative. Please note that failure to state your preference will result in delays in accepting your manuscript for publication.

Cover suggestions

As you prepare your final files we encourage you to consider whether you have any images or illustrations that may be appropriate for use on the cover of Nature Ecology & Evolution.

Nature Ecology & Evolution has now transitioned to a unified Rights Collection system which will allow our Author Services team to quickly and easily collect the rights and permissions required to publish your work. Approximately 10 days after your paper is formally accepted, you will receive an email in providing you with a link to complete the grant of rights. If your paper is eligible for Open Access, our Author Services team will also be in touch regarding any additional information that may be required to arrange payment for your article.

Please note that *Nature Ecology & Evolution* is a Transformative Journal (TJ). Authors may publish their research with us through the traditional subscription access route or make their paper immediately open access through payment of an article-processing charge (APC). Authors will not be required to make a final decision about access to their article until it has been accepted. [Find out more about Transformative Journals](https://www.springernature.com/gp/open-research/transformative-journals)

Authors may need to take specific actions to achieve [compliance](https://www.springernature.com/gp/open-research/funding/policy-compliance-faqs) with funder and institutional open access mandates. If your research is supported by a funder that requires immediate open access (e.g. according to [a](https://www.springernature.com/gp/open-research/funding/policy-compliance-faqs)

26[Plan S principles](https://www.springernature.com/gp/open-research/plan-s-compliance)) then you should select the gold OA route, and we will direct you to the compliant route where possible. For authors selecting the subscription publication route, the journal's standard licensing terms will need to be accepted, including <https://www.nature.com/nature-portfolio/editorial-policies/self-archiving-and-license-to-publish>. Those licensing terms will supersede any other terms that the author or any third party may assert apply to any version of the manuscript.

[REDACTED]

[REDACTED]

Reviewer #1:

Remarks to the Author:

The authors have addressed all of my comments and given valid responses and I do not have anything further. Based on the responses to mine and the other reviewer's comments I would be happy to see this article published.

Reviewer #2:

Remarks to the Author:

I appreciate that the authors have worked hard to revise and improve the manuscript according to the reviewers' comments. All the main concerns and suggestions that I raised have been successfully addressed, implemented or clarified. This reviewer thanks the authors for their detailed replies to those points and for adjusting the manuscript following most of the suggestions. The additional analyses and extra supplementary material included in this revised version also reinforce some key points and enhance the quality of the paper. Overall, a neat revision of an already good manuscript that I consider it is ready to be published. Congratulations for an excellent work and a valuable contribution to the field.

Final Decision Letter:

3rd July 2023

Dear Dr Sethi,

We are pleased to inform you that your Brief Communication entitled "Limits to the accurate and generalisable use of soundscapes to monitor biodiversity", has now been accepted for publication in Nature Ecology & Evolution.

Over the next few weeks, your paper will be copyedited to ensure that it conforms to Nature Ecology and Evolution style. Once your paper is typeset, you will receive an email with a link to choose the appropriate publishing options for your paper and our Author Services team will be in touch regarding any additional information that may be required

You will not receive your proofs until the publishing agreement has been received through our system

Due to the importance of these deadlines, we ask you please us know now whether you will be difficult to contact over the next month. If this is the case, we ask you provide us with the contact information (email, phone and fax) of someone who will be able to check the proofs on your behalf, and who will be available to address any last-minute problems . Once your paper has been scheduled for online publication, the Nature press office will be in touch to confirm the details.

Acceptance of your manuscript is conditional on all authors' agreement with our publication policies (see www.nature.com/authors/policies/index.html). In particular your manuscript must not be published elsewhere and there must be no announcement of the work to any media outlet until the publication date (the day on which it is uploaded onto our web site).

Please note that *Nature Ecology & Evolution* is a Transformative Journal (TJ). Authors may publish their research with us through the traditional subscription access route or make their paper immediately open access through payment of an article-processing charge (APC). Authors will not be required to make a final decision about access to their article until it has been accepted. [Find out more about Transformative Journals](https://www.springernature.com/gp/open-research/transformative-journals)

Authors may need to take specific actions to achieve [compliance](https://www.springernature.com/gp/open-research/funding/policy-compliance-faqs) with funder and institutional open access mandates. If your research is supported by a funder that requires immediate open access (e.g. according to [Plan S principles](https://www.springernature.com/gp/open-research/plan-s-compliance))

28then you should select the gold OA route, and we will direct you to the compliant route where possible. For authors selecting the subscription publication route, the journal's standard licensing terms will need to be accepted, including <https://www.nature.com/nature-portfolio/editorial-policies/self-archiving-and-license-to-publish>. Those licensing terms will supersede any other terms that the author or any third party may assert apply to any version of the manuscript.

We welcome the submission of potential cover material (including a short caption of around 40 words) related to your manuscript; suggestions should be sent to Nature Ecology & Evolution as electronic files (the image should be 300 dpi at 210 x 297 mm in either TIFF or JPEG format). Please note that such pictures should be selected more for their aesthetic appeal than for their scientific content, and that colour images work better than black and white or grayscale images. Please do not try to design a cover with the Nature Ecology & Evolution logo etc., and please do not submit composites of images related to your work. I am sure you will understand that we cannot make any promise as to whether any of your suggestions might be selected for the cover of the journal.

You can generate the link yourself when you receive your article DOI by entering it here: <http://authors.springernature.com/share>.

Yours sincerely,

Patrick Goymer, DPhil

29Chief Editor
Nature Ecology & Evolution

P.S. Click on the following link if you would like to recommend Nature Ecology & Evolution to your librarian <http://www.nature.com/subscriptions/recommend.html#forms>

** Visit the Springer Nature Editorial and Publishing website at http://editorial-jobs.springernature.com?utm_source=ejp_NEcoE_email&utm_medium=ejp_NEcoE_email&utm_campaign=ejp_NEcoE for more information about our career opportunities. If you have any questions please click [here](mailto:editorial.publishing.jobs@springernature.com). **